# Generating with Fairness: A Modality-Diffused Counterfactual Framework for Incomplete Multimodal Recommendations

## ABSTRACT

Incomplete scenario is a prevalent, practical, yet challenging setting in Multimodal Recommendations (MMRec), where some item modalities are missing due to various factors. Recently, a few efforts have sought to improve the recommendation accuracy by exploring generic structures from incomplete data. However, two significant gaps persist: 1) the difficulty in accurately generating missing data due to the limited ability to capture modality distributions; and 2) the critical but overlooked visibility bias, where items with missing modalities are more likely to be disregarded due to the prioritization of items' multimodal data over user preference alignment. This bias raises serious concerns about the fair treatment of items. To bridge these two gaps, we propose a novel Modality-Diffused Counterfactual (MoDiCF) framework for incomplete multimodal recommendations. MoDiCF features two key modules: a novel modality-diffused data completion module and a new counterfactual multimodal recommendation module. The former, equipped with a particularly designed multimodal generative framework, accurately generates and iteratively refines missing data from learned modality-specific distribution spaces. The latter, grounded in the causal perspective, effectively mitigates the negative causal effects of visibility bias and thus assures fairness in recommendations. Both modules work collaboratively to address the two aforementioneds significant gaps for generating more accurate and fair results. Extensive experiments on three real-world datasets demonstrate the superior performance of MoDiCF in terms of both recommendation accuracy and fairness. The code and processed datasets are released at https://anonymous.4open.science/r/MoDiCF-EEF5.

## CCS CONCEPTS

• **Information systems** → **Retrieval tasks and goals**.

## KEYWORDS

Multimodal recommendations, Missing modalities, Visibility bias

**ACM Reference Format:**
Anonymous Author(s). 2018. Generating with Fairness: A Modality-Diffused Counterfactual Framework for Incomplete Multimodal Recommendations. In *Proceedings of Make sure to enter the correct conference title from your rights confirmation emai (Conference acronym 'XX).* ACM, New York, NY, USA, 12 pages. https://doi.org/XXXXXXX.XXXXXXX

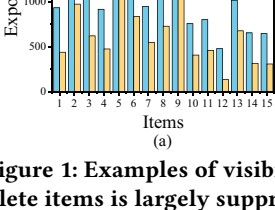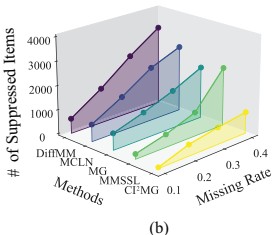

**Figure 1: Examples of visibility bias. (a): Exposure of incomplete items is largely suppressed compared to complete scenarios. (b): Suppression aggravates as missing rates increase.**

## 1 INTRODUCTION

Multimodal data has recently shown massive growth across various multimedia applications, including social media, news and e-commerce platforms [38]. To address information overload and help users discover interesting items, Multimodal Recommendations (MMRec) [5, 22, 24, 58, 59] have emerged as a prevalent research topic. MMRec integrates items' diverse modalities (e.g., item images, review texts) into user preference modeling, thereby delivering more accurate recommendations. However, most MMRec methods heavily rely on the assumption that all modalities are complete (i.e., fully observed) [1], which doesn't always hold in the real world. Factors like privacy concerns, data sparsity, and acquisition costs often result in incomplete items, with some modalities missing. This poses a significant challenge to existing methods, as they struggle to exploit useful information from limited data.

In recent years, a few efforts [1, 20, 30, 41] have been made to handle incomplete data in MMRec. Among them, two primary strategies have been researched: 1) data completion, which aims to recover missing data using different imputation methods, such as autoencoder [41], feature propagation [30], clustering-based hypergraph convolution and cross-modal transport [20], and 2) modality-invariant learning [1], which aims to learn inherent content preferences that is invariant to modality missing.

Despite achieving promising accuracy improvements, we identify two major gaps confronted by these existing methods. **Gap 1**: *they often struggle to accurately recover multimodal missing data due to their limited capability in capturing modality-specific distributions.* Existing methods typically focus on learning generic data structures without explicitly exploring data distributions within each modality and their differences across modalities [4]. This is particularly crucial in incomplete MMRec, where severe information loss disrupts data structure learning [9], and, moreover, high heterogeneity commonly exists across modalities [19], making the learning process more challenging. For instance, the appearance image and description texts of an item may convey similar semantics but belong to distinct distributions. Failing to accurately capture such modality-specific distributions can greatly diminish data completion quality, leading to suboptimal recommendation performance [22]. Recently,

diffusion models [2, 13, 44] have gained widespread attention for their ability to capture complex data distributions. Though several diffusion-based recommendation methods have been developed [27, 54], they are not applicable in incomplete MMRec due to their unimodal design and inability to handle incomplete data. Thus, there is an urgent need for a specifically devised diffusion module tailored for modality completion in MMRec to fill this gap.

**Gap 2**: *they largely overlook a critical fairness issue, namely the **visibility bias of items***. This bias refers to the phenomenon that items with missing modalities are more likely to be overlooked or receive less exposure from recommenders, regardless of their genuine alignment with users' preferences. As an example in Figure 1 (a), the exposure of items in the Baby dataset by an existing MMRec method, MMSSL [46], significantly decreases in incomplete scenarios, confirming the existence of visibility bias. Moreover, Figure 1 (b) shows that as the data missing rate increases, the number of suppressed items rises rapidly, indicating a growing severity of visibility bias. This bias often leads to the unfair treatment of items and, consequently, inaccurate and biased recommendation results. For example, on e-commerce platforms, niche or artisan products may have limited visual content, which reduces their visibility and sales opportunities. This not only affects the sellers but also deprives the users of potentially better-suited products. Such critical yet overlooked bias demands careful attention and a targeted solution. However, addressing it is a non-trivial task that requires a special focus on the causal impacts [31] of incomplete multimodal content on recommendation outcomes. Most existing debiasing methods [35, 45] overlook incomplete scenarios and fail to identify the unique cause of visibility bias. Therefore, they are not ready to address this particular problem.

To this end, in this paper, we propose a novel **Mo**dality-**Di**ffused **C**ounter-**F**actual (**MoDiCF**) framework for incomplete MMRec to address the two aforementioned significant gaps. MoDiCF features the following two well-designed modules. 1) **Modality-diffused data completion module.** Built within a generative framework, this module excels at capturing and generating missing modalities from complex modality-specific distributions. To effectively harness cross-modal correlations and inject them into the diffusion process, we introduce modality-aware conditioning. It provides complementary modality information to enhance modality-specific distribution learning and facilitate cross-modal alignment. Additionally, we devise a novel iterative refinement strategy to continuously update and improve the generated missing data, further enhancing the quality of data completion. 2) **Counterfactual multimodal recommendation module.** In this module, we perform counterfactual adjustments on the MMRec process to mitigate the visibility bias and deliver accurate and fair recommendations. We first examine the visibility bias problem through a causal lens and identify its root cause as an over-reliance on multimodal content rather than user-item preference alignment. Based on this insight, we design this module with two sub-modules, a multimodal recommender and a multimodal item predictor. The former serves as a standard MMRec model, which leverages completed multimodal data to predict user-item matching scores. The latter focuses on estimating item ranking scores based on multimodal data only, thus disentangling the effects of visibility bias – specifically, the tendency to suppress items simply due to incomplete multimodal data. By integrating

the results from these two sub-modules with counterfactual inference, visibility bias can be effectively mitigated. These two modules work collaboratively to improve both recommendation accuracy and fairness of incomplete MMRec.

In summary, the main contributions of this work are as follows:

- We propose a novel modality-diffused counterfactual framework for incomplete MMRec to generate accurate and fair recommendations by effectively addressing data incompleteness and visibility bias issues.
- We devise a novel modality-aware diffusion module to accurately generate data for missing modalities, guided by multimodal conditions. This module excels at capturing complex data distributions and ensuring cross-modal consistency compared to existing incomplete MMRec methods.
- We design a new counterfactual multimodal recommendation module to effectively identify and mitigate visibility bias through counterfactual inference.

Our proposed framework could be easily instantiated into various specific models by taking different MMRec models as the multimodal recommender, see Section F for more details. In this paper, we instantiate it as one specific model. Evaluations on three real-world datasets show that our model significantly outperforms state-of-the-art methods in both recommendation accuracy and fairness.

## 2 RELATED WORK

**Multimodal Recommender Systems.** Multimodal recommender systems [3] have garnered significant attention. The key challenge lies in how to effectively utilize diverse multimodal content and explore modality-specific user preferences. Early attempts, such as VBPR [11], address this by integrating multimedia features into matrix decomposition. More recently, Graph Neural Networks (GNNs) have been extensively researched [10, 47, 48, 56] to encode multimodal content into latent features. Representative works incorporate attention mechanisms [39], self-supervised learning [46] and modality-aware contrastive learning [21, 46] for enhanced performance. However, most of them struggle with incomplete data. Efforts to address this issue mainly include autoencoder-based methods [41], clustering-based hypergraph convolution methods [20] and feature propagation methods [30]. Alternatively, some others have focused on learning users' inherent preferences [1], which remain invariant despite data missing. Nonetheless, they still face two significant gaps: 1) the difficulty in accurately generating missing data, and 2) the inability to mitigate visibility bias, a critical yet commonly overlooked fairness issue in practice.

**Diffusion-based Recommender Systems.** Diffusion models have made substantial progress in data reconstruction [62], generation [49] and denoising [13]. This promotes their integration into various recommendation tasks such as outfit recommendations [53], point-of-interest recommendations [26, 32] and sequential recommendations [23, 27, 28, 50, 54]. In the realm of MMRec [55], Ma *et al.* [29] employ a standard diffusion process to enhance item representations by integrating multimodal knowledge and reducing noise. Similarly, Jiang *et al.* [14] use a graph diffusion model to improve multimodal alignment and enhance performance by embedding multimodal information into the user-item graph. Despite these advancements, there remains a lack of a well-designed diffusion

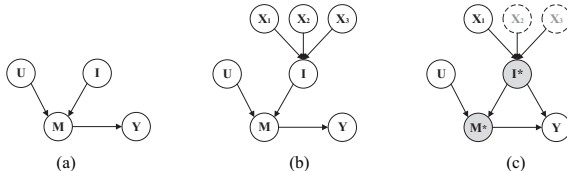

Figure 2: Causal graph for (a) RS, (b) MMRec and (c) incomplete MMRec. U: user features, I: item features, M: matching features, Y: ranking scores, $X_i$: multimodal data. The dashed lines indicate the missing modalities.

module to capture multimodal correlations and handle incomplete data in MMRec. Our work aims to address this crucial shortfall.

**Counterfactual Debiasing in Recommender Systems.** Counterfactual inference [31], as one of the powerful causal techniques, is used to explore and intervene in the causal relationships of underlying biases in hypothetical scenarios. This approach has demonstrated significant value in building trustworthy Recommender Systems (RSs) [6, 16], particularly for debiasing. Pioneering research in this direction includes efforts to tackle popularity bias [45], attribute bias [18] and filter bubble bias [7, 43]. In MMRec, a few counterfactual methods have been proposed recently. For instance, Shang *et al.* [35] address the bias stemming from the dominance of modalities, while Li *et al.* [17] reduce spurious correlations from user-uninteracted items. However, a critical visibility bias remains underexplored. This gap calls for an in-depth causal analysis of how incompleteness potentially impacts MMRec mechanisms, a need that has yet to be addressed.

## 3 PROBLEM FORMULATION

Let $\mathcal{U}$ denote a set of $N_U$ users, $\mathcal{I}$ denote a set of $N_I$ items, we represent the user-item interaction matrix as $\mathbf{Y} \in \mathbb{R}^{N_U \times N_I}$. We have $\mathbf{Y}_{u,i} = 1$ if user $u \in \mathcal{U}$ has interacted with item $i \in \mathcal{I}$, and $\mathbf{Y}_{u,i} = 0$ otherwise. In the multimodal setting, each item has $M$ modalities, represented as $\mathcal{X}^m = \{\mathbf{x}_i^m\}_{i=1}^{N_I}$, where $\mathbf{x}_i^m \in \mathbb{R}^{d_m}$ is the $m$-th modality with a dimension $d_m$ and $m \in [1, M]$. Following general practice [36, 52], we reduce $d_m$ to 128 via singular value decomposition for improved efficiency. The goal of MMRec is to predict the matching score $\mathbf{Y}_{u,i}$ between $u$ and $i$. Beyond this, we consider two significant issues: 1) missing modalities, and 2) visibility bias.

**Missing Modalities**. Different from complete items, which have all modalities present, items in practice are often associated with missing modalities and are referred to as incomplete items. Here, we consider a generalized case where each item may randomly have $\tilde{m} \in [0, M-1]$ modalities missing. This means the degree of incompleteness varies across items, but at least one modality is observed per item. We define an indicator matrix $\mathbf{E} \in \{0, 1\}^{N_I \times M}$ to record incompleteness, where $\mathbf{E}_{i,m} = 0$ indicates missing modality and $\mathbf{E}_{i,m} = 1$ otherwise. Thus, we have the observed set and missing set $\mathcal{E}_{\text{ob}}^m = \{i | \mathbf{E}_{i,m} = 1\}$ and $\mathcal{E}_{\text{ms}}^m = \{i | \mathbf{E}_{i,m} = 0\}$. Accordingly, multimodal data can be split into $\mathcal{X}_{\text{ob}}^m = \{\mathbf{x}_i^m | i \in \mathcal{E}_{\text{ob}}^m\}$ and $\mathcal{X}_{\text{ms}}^m = \{\mathbf{x}_i^m | i \in \mathcal{E}_{\text{ms}}^m\}$ with $\mathcal{X}^m = \mathcal{X}_{\text{ob}}^m \cup \mathcal{X}_{\text{ms}}^m$. Note that, for $\mathcal{X}_{\text{ms}}^m$, we initialize them with $\mathbf{0} \in \mathbb{R}^{d_m}$. One of our main goals is to accurately recover these missing modalities for accurate recommendations.

**Visibility Bias**. To analyze visibility bias from a fundamental causal perspective, we first formulate the MMRec problem as a

structural causal model [31] in Figure 2 (b). Three causal relations are observed: 1) $\{\mathbf{X}_i\}_{i=1}^3 \rightarrow \mathbf{I}$ indicates the integration of multimodal data into item features; 2) $\mathbf{U}, \mathbf{I} \rightarrow \mathbf{M}$ indicates the learning of user-item preference matching; and 3) $\mathbf{M} \rightarrow \mathbf{Y}$ represents the prediction of ranking scores based on matching features. While in an incomplete scenario of Figure 2 (c), an additional causal path $\mathbf{I}^* \rightarrow \mathbf{Y}$ emerges. This forms a shortcut to generate recommendations, instigating the model to prioritize items' multimodal data over user preference alignment. Hence, incomplete items can hardly compete with complete ones, leading to reduced exposure and visibility bias. We aim to solve it through counterfactual inference.

## 4 THE MODICF FRAMEWORK

The proposed MoDiCF framework, as shown in Figure 3, consists of two key modules: a Modality-Diffused Data Completion (MDDC) module and a CounterFactual Multimodal Recommendation (CFMR) Module. For incomplete items, the framework first maps the observed data into latent spaces and learns modality-specific distributions through a diffusion process. Particularly, modality-aware conditions are injected to guide the learning and ensure cross-modal consistency. Then, missing modalities can be generated from the learned distributions and refined iteratively. Based on the completed multimodal data, the CFMR module is devised to address visibility bias with two sub-modules, i.e., the multimodal recommender and the item predictor. By drawing on causal insights, we can effectively disentangle the negative effects of visibility bias using the item predictor and eliminate them from the multimodal recommender's results. These modules are trained collaboratively to deliver more accurate and fair recommendations.

### 4.1 Modality-Diffused Data Completion Module

This module generally follows the prevalent denoising diffusion probabilistic paradigm [13] for high compatibility and extensibility. However, it is tailored for multimodal data completion with two specific designs: 1) diffusion with modality-aware conditioning to incorporate cross-modal correlations; and 2) iterative refinement to progressively enhance the generated data and reduce ambiguity from missing information.

*4.1.1 Diffusion with Modality-aware Conditioning.* As shown in Figure 3, we consider the diffusion process independently for different modalities, as they follow distinct distributions. For the $m$-th modality, we first encode the observed data $\mathcal{X}_{\text{ob}}^m$ into a latent space using a latent encoder $E_{\text{diff}}^m$ and thus obtain a set of latent representations $\mathcal{V}_{\text{ob}}^m$. A diffusion process is then trained in this latent space to capture modality-specific distributions through a series of forward and reverse steps. Starting with $\mathbf{v}_{\text{ob},0}^m$, the forward process transforms the original modality distribution into Gaussian noise over $T$ steps (we set $T = 1000$ following [13]), formulated as:

$$q(\mathbf{v}_{\text{ob},1:T}^m | \mathbf{v}_{\text{ob},0}^m) = \prod_{t=1}^{T} q(\mathbf{v}_{\text{ob},t}^m | \mathbf{v}_{\text{ob},t-1}^m),$$
$$q(\mathbf{v}_{\text{ob},t}^m | \mathbf{v}_{\text{ob},t-1}^m) = \mathcal{N}(\mathbf{v}_{\text{ob},t}^m; \sqrt{1-\beta_t} \mathbf{v}_{\text{ob},t-1}^m, \beta_t \mathbf{I}), \quad (1)$$

where $t \in [1, T]$ and $\beta_1, \cdots, \beta_T$ are a predefined variance schedule. In the reverse process, the goal is to recover $\mathbf{v}_{\text{ob},0}^m$ from $p_\theta(\mathbf{v}_{\text{ob},T}^m)$. However, due to incompleteness, the lack of sufficient observable

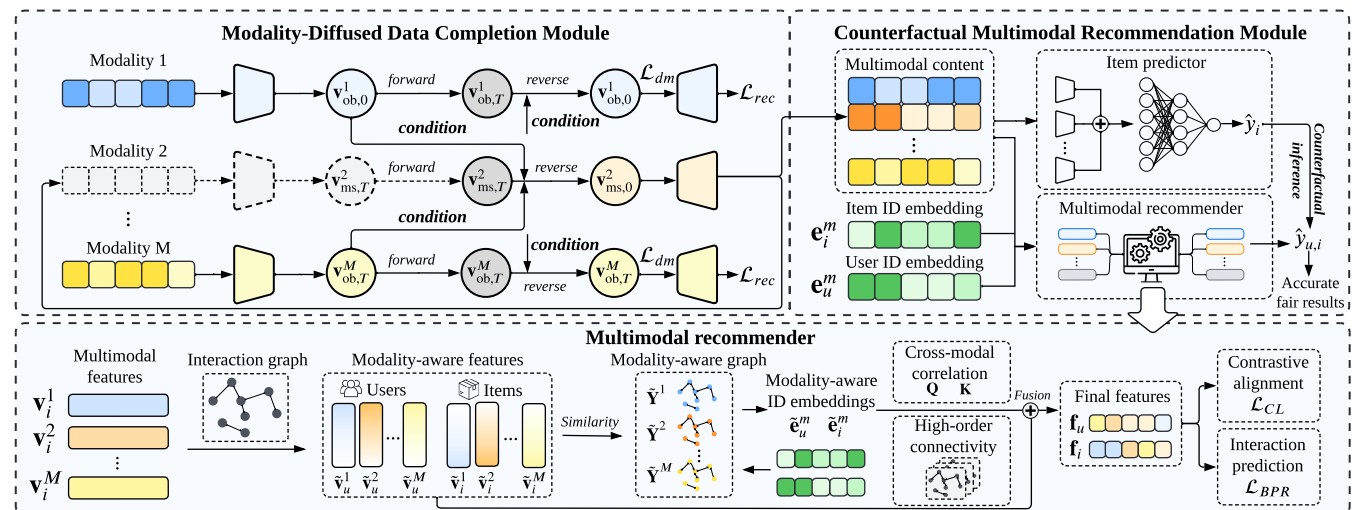

**Figure 3: The MoDiCF framework comprises two key modules: a modality-diffused data completion module and a counterfactual multimodal recommendation module. The former is devised to generate data for missing modalities from captured modality-specific distributions, guided by modality-aware conditions. The latter aims to address visibility bias from a causal perspective.**

samples can lead to severe information loss. More importantly, unimodal diffusion often fails to achieve semantic consistency across different item modalities. To address these, we introduce a novel **modality-aware condition mechanism** to guide the diffusion. First, features from other modalities of items in $\mathcal{E}_{ob}^m$ are gathered as $\mathcal{V}_{cn}^m = \{\mathbf{v}_i^j | i \in \mathcal{E}_{ob}^m, j = 1, \cdots, M, j \neq m\}$. Then, they are integrated using a fusion network $f_{cn}$, producing modality-aware conditions $\mathbf{v}_{cn,0}^m$. Instead of directly learning $p_\theta(\mathbf{v}_{ob,t-1}^m | \mathbf{v}_{ob,t}^m)$, we present a modality-conditioned model $\theta_{cn}$ to take complementary modality information into account, formulating the reverse process as:

$$p_{\theta_{cn}}(\mathbf{v}_{ob,t-1}^m | \mathbf{v}_{ob,t}^m, \mathbf{v}_{cn,t}^m) = \mathcal{N}(\mathbf{v}_{ob,t-1}^m; \mu_{\theta_{cn}}(\mathbf{v}_{ob,t}^m, \mathbf{v}_{cn,t}^m, t), \sigma_t^2 \mathbf{I}), \quad (2)$$

where $\mathbf{v}_{cn,t}^m$ is the condition at step $t$, and $\sigma_t^2$ is the variance [13]. The mean $\mu_{\theta_{cn}}(\mathbf{v}_{ob,t}^m, \mathbf{v}_{cn,t}^m, t)$ is computed using the following parameterization strategy:

$$\mu_{\theta_{cn}}(\mathbf{v}_{ob,t}^m, \mathbf{v}_{cn,t}^m, t) = \frac{1}{\sqrt{\alpha_t}} \left( \mathbf{v}_{ob,t}^m - \frac{\beta_t}{\sqrt{1-\bar{\alpha}_t}} \epsilon_{cn}(\mathbf{v}_{ob,t}^m, \mathbf{v}_{cn,t}^m, t) \right), \quad (3)$$

where $\alpha_t = 1 - \beta_t$, $\bar{\alpha}_t = \prod_{i=1}^t \alpha_i$, and $\epsilon_{cn}$ is a neural network based on the U-Net architecture [13, 34] to predict $\epsilon \sim \mathcal{N}(\mathbf{0}, \mathbf{I})$. To reinforce the integration of conditions, we perform attention-based [40, 44] condition fusion in $\epsilon_{cn}$. Assume the representations of $\mathbf{v}_{ob,t}^m$ and $\mathbf{v}_{cn,t}^m$ in the $l$-th layer of $\epsilon_{cn}$ are $\mathbf{v}_{ob,t}^{m(l)}$ and $\mathbf{v}_{cn,t}^{m(l)}$, we have:

$$\mathbf{v}_{fuse,t}^{m(l)} = \text{softmax}\left( \frac{1}{\sqrt{d^{(l)}}} (\mathbf{v}_{ob,t}^{m(l)} \mathbf{W}_Q^{(l)})^\top (\mathbf{v}_{cn,t}^{m(l)} \mathbf{W}_K^{(l)}) \right) (\mathbf{v}_{cn,t}^{m(l)} \mathbf{W}_V^{(l)}), \quad (4)$$

where $\mathbf{W}_Q^{(l)}$, $\mathbf{W}_K^{(l)}$ and $\mathbf{W}_V^{(l)}$ are learnable parameters, and $d^{(l)}$ is the dimension of the $l$-th layer. The fused representation $\mathbf{v}_{fuse,t}^{m(l)}$ is then forwarded to the next layer.

After this modality-aware diffusion process, a latent decoder $D_{diff}^m$ is applied to map latent representations back to the original modality space. The loss objective for this module $\mathcal{L}_{diff}$ is composed of two terms: the diffusion loss $\mathcal{L}_{dm}$ and the reconstruction loss

$\mathcal{L}_{rec}$, balanced by $\alpha_1$ (the value is set as Table 4 in the appendix). The loss objective is defined as:

$$\mathcal{L}_{diff} = \mathcal{L}_{dm} + \alpha_1 \mathcal{L}_{rec}, \quad (5)$$

$$\mathcal{L}_{dm} = \mathbb{E}_{t,\epsilon} \left[ \|\epsilon - \epsilon_{cn}(\mathbf{v}_{ob,t}^m, \mathbf{v}_{cn,t}^m, t)\|_2^2 \right], \quad (6)$$

$$\mathcal{L}_{rec} = \mathbb{E}_{\mathbf{x}_{ob}^m \sim \mathcal{X}_{ob}^m} \left[ \|D_{diff}^m(E_{diff}^m(\mathbf{x}_{ob}^m)) - \mathbf{x}_{ob}^m\|_2^2 \right]. \quad (7)$$

*4.1.2 Iterative Refinement.* Besides the modality-aware conditions, another notable difference between our method and existing diffusion models [1, 13, 27, 53] is the iterative refinement strategy we propose. This allows us to leverage both complete and incomplete items during training, and progressively refine the quality of generated data. We begin by initializing missing modalities $\mathcal{X}_{ms}^m$ with zero vectors, and then apply the modality-aware diffusion process to learn modality-specific distributions. Next, we aim to perform diffusion generation to update incomplete modality data. Specifically, for the $m$-th modality, we draw Gaussian noise $\mathbf{v}_{ms,T_s}^m$ from $\mathcal{N}(\mathbf{0}, \mathbf{I})$. Similar to the reverse process, we gather representations from other modalities and inject modality-aware conditions $\mathbf{v}_{cn,T_s}^m$ into the generation process for guidance. For each sampling step $t = T_s, \cdots, 1$, we use the following generation process to recover missing representations:

$$\mathbf{v}_{ms,t-1}^m = \frac{1}{\sqrt{\alpha_t}} \left( \mathbf{v}_{ms,t}^m - \frac{\beta_t}{\sqrt{1-\bar{\alpha}_t}} \epsilon_{cn}(\mathbf{v}_{ms,t}^m, \mathbf{v}_{cn,t}^m, t) \right) + \sigma_t \mathbf{z}, \quad (8)$$

where $\mathbf{z} \sim \mathcal{N}(\mathbf{0}, \mathbf{I})$ if $t > 1$, and $\mathbf{z} = \mathbf{0}$ otherwise. To accelerate the generation process, we apply an efficient sampling strategy from [37]. The generated missing modality $\hat{\mathbf{x}}_{ms}^m$ is then obtained by decoding $\mathbf{v}_{ms,0}^m$ with the latent decoder $D_{diff}^m$. By replacing each $\mathbf{x}_{ms}^m$ with $\hat{\mathbf{x}}_{ms}^m$ and repeating these steps throughout the training, newly recovered data continuously participate in the diffusion process and effectively reduce the ambiguity due to missing information. In this way, the quality of modality completion is progressively refined for improving MMRec performance.

## 4.2 Counterfactual Multimodal Recommendation Module

Building upon the well-designed MDDC module, we can effectively generate complete multimodal data for recommendations. However, visibility bias remains unaddressed, as it arises from inherent mechanisms within incomplete MMRec. Since data-driven methods usually fall short in identifying the root causes of underlying biases [16], we shift to a causal perspective for resolution. In this module, we integrate counterfactual inference into MMRec process to deliver accurate and fair recommendations.

Following counterfactual inference, we aim to explore this hypothetical question: *what would the ranking scores* ($\mathbf{Y}$) *be if items* ($\mathbf{I}$) *were incomplete and visibility bias emerged?* This suggests us to measure the causal effects on the outcome variable $\mathbf{Y}$ by changing the treatment variable $\mathbf{I}$ from the current complete scenario, $\mathbf{I} = i$, to a hypothetical scenario, $\mathbf{I} = i^*$, where incompleteness and visibility bias exist. This is known as the Total Effect (TE) [31, 45]:

$$TE = \mathbf{Y}_{i,\mathbf{M}_i} - \mathbf{Y}_{i^*,\mathbf{M}_{i^*}}, \qquad (9)$$

where $\mathbf{Y}_{i,\mathbf{M}_i}$ and $\mathbf{Y}_{i^*,\mathbf{M}_{i^*}}$ are respectively the values of ranking scores under two different scenarios: $\mathbf{I} = i$ and $\mathbf{I} = i^*$. However, as we may recall from Section 3, both the direct path $\mathbf{I} \rightarrow \mathbf{Y}$ and the indirect path $\mathbf{I} \rightarrow \mathbf{M} \rightarrow \mathbf{Y}$ contribute to TE. The former is the root cause of the visibility bias while the latter is desired for accurate and fair recommendations. To disentangle visibility bias, we decompose TE into the Natural Direct Effect (NDE) and the Total Indirect Effect (TIE), capturing the effects of direct and indirect paths, respectively. Therefore, addressing visibility bias is now formulated as eliminating NDE from TE and focusing on the learning of TIE:

$$NDE = \mathbf{Y}_{i,\mathbf{M}_{i^*}} - \mathbf{Y}_{i^*,\mathbf{M}_{i^*}}, \ TIE = TE - NDE = \mathbf{Y}_{i,\mathbf{M}_i} - \mathbf{Y}_{i,\mathbf{M}_{i^*}}. \ (10)$$

Based on this insight, we design a novel CFMR module with two sub-modules, namely a multimodal recommender and an item predictor. The former predicts ranking scores based on user-item matching features while the latter only relies on multimodal data to directly measure the negative effects of visibility bias. Their predictions are then aggregated following Eq. 10 for debiasing.

*4.2.1 Multimodal Recommender.* Motivated by the recent success [46, 48, 61] of graph representation learning in MMRec, we present a multimodal recommender which incorporates multimodal features into user-item graph learning. Specifically, we first construct a user-item graph $\mathcal{G} = \{(u,i)|u \in \mathcal{U}, i \in \mathcal{I}\}$ by taking users and items as nodes, and user-item interactions $\mathbf{Y}_{u,i}$ as edges. Based on the completed multimodal data $\hat{\mathcal{X}}^m$, we extract latent representations $\mathcal{V}^m = \{\mathbf{v}_1^m, \cdots, \mathbf{v}_{N_I}^m\}$ for each modality using a latent encoder, where $\mathbf{v}_i^m \in \mathbb{R}^d$, $d$ is the dimension of the latent space. For each modality, we derive modality-aware user and item features from latent representations via:

$$\tilde{\mathbf{v}}_u^m = \sum_{a \in \mathcal{N}_u} \mathbf{v}_a^m / \sqrt{|\mathcal{N}_u|}, \quad \tilde{\mathbf{v}}_i^m = \sum_{b \in \mathcal{N}_i} \mathbf{v}_b^m / \sqrt{|\mathcal{N}_i|}, \qquad (11)$$

where $\mathcal{N}_u$ and $\mathcal{N}_i$ are the neighborhood sets of users and items in $\mathcal{G}$. Then, we learn a modality-aware user-item interaction matrix $\mathbf{Y}^m$ for the $m$-th modality, by calculating $\mathbf{Y}_{u,i}^m = \text{sim}(\tilde{\mathbf{v}}_u^m, \tilde{\mathbf{v}}_i^m)$, where $\text{sim}(\cdot, \cdot)$ is the cosine similarity. Based on this, we perform information aggregation for user and item ID embeddings $\mathbf{e}_u \in \mathbf{E}_U$

and $\mathbf{e}_i \in \mathbf{E}_I$ with the following:

$$\mathbf{e}_u^m = \sum_{a \in \mathcal{N}_u^m} \mathbf{e}_a / \sqrt{|\mathcal{N}_u^m|}, \quad \mathbf{e}_i^m = \sum_{b \in \mathcal{N}_i^m} \mathbf{e}_b / \sqrt{|\mathcal{N}_i^m|}, \qquad (12)$$

where $\mathcal{N}_u^m$ and $\mathcal{N}_i^m$ are the neighborhood sets derived from $\mathbf{Y}^m$.

To further improve representation learning, we perform the multi-head self-attention mechanism [40] to learn cross-modal correlations. Given $H$ attention heads, the attention mechanism can be formulated as:

$$\bar{\mathbf{e}}_u^m = \sum_{n=1}^M \overset{H}{\underset{h=1}{\|}} \text{softmax}\left(\frac{1}{\sqrt{d/H}}(\mathbf{e}_u^m \mathbf{W}_h^Q)^\top \cdot (\mathbf{e}_u^n \mathbf{W}_h^K)\right) \cdot \mathbf{e}_u^n, \quad (13)$$

where $\mathbf{W}_h^Q$, $\mathbf{W}_h^K$ are learnable parameters of query and key transformations. Then, multimodal embeddings are integrated with the mean-pooling strategy $\bar{\mathbf{e}}_u = \sum_{m=1}^M \bar{\mathbf{e}}_u^m / M$. The corresponding item embeddings can be obtained analogously. Meanwhile, to capture high-order connectivity patterns, we perform recursive message passing to refine the embeddings:

$$\hat{\mathbf{E}}_U^{(l+1)} = \mathbf{Y} \cdot \hat{\mathbf{E}}_I^{(l)}, \qquad (14)$$

where $\hat{\mathbf{E}}_U$ is initialized as $\hat{\mathbf{E}}_U^0 = \mathbf{E}_U + \eta \bar{\mathbf{E}}_U / \|\bar{\mathbf{E}}_U\|_2^2$, $\eta$ is a hyperparameter with its value in Table 4 and $\bar{\mathbf{E}}_U = [\bar{\mathbf{e}}_1, \cdots, \bar{\mathbf{e}}_{N_U}]$. After $L$ layers of updating, we integrate the embeddings of each layer through $\hat{\mathbf{E}}_u = \sum_{l=1}^L \hat{\mathbf{E}}_U^{(l)} / L$. The resulting user embeddings are obtained by $\mathbf{f}_u = \hat{\mathbf{e}}_u + \delta \sum_{m=1}^M \tilde{\mathbf{v}}_u^m / \|\tilde{\mathbf{v}}_u^m\|_2$, where $\delta$ is the hyperparameter of fusion weight, whose value is specified in Table 4. Item embeddings can be computed analogously. Consequently, the interaction between user $u$ and item $i$ is predicted through $\hat{y}_{u,i} = \mathbf{f}_u^\top \mathbf{f}_i$.

The training objective of multimodal recommender includes two parts: cross-modal contrastive learning and interaction prediction. The former injects additional self-supervised signal [46] into MMRec and is trained with the following loss function:

$$\mathcal{L}_{\text{CL}} = -\sum_{m=1}^M \sum_{u=1}^{N_U} \log \frac{\exp(\text{sim}(\mathbf{f}_u, \mathbf{e}_u^m))}{\sum_{v=1}^{N_U} (\exp(\text{sim}(\mathbf{f}_v, \mathbf{e}_u^m)) + \exp(\text{sim}(\mathbf{e}_v^m, \mathbf{e}_u^m)))}. \quad (15)$$

The latter focuses on the prediction of user-item interactions, which is supervised by the Bayesian Personalized Ranking (BPR) loss [33]:

$$\mathcal{L}_{\text{BPR}_{u,i}} = f_{\text{BPR}}(\hat{y}_{u,i_p}, \hat{y}_{u,i_n}) = -\sum_{(u,i_p,i_n)}^{|\mathcal{A}|} \log\left(\text{sigm}(\hat{y}_{u,i_p} - \hat{y}_{u,i_n})\right), \quad (16)$$

where $i_p$ and $i_n$ are positive and negative items for $u$, and $\text{sigm}(\cdot)$ is the sigmoid function.

*4.2.2 Item Predictor.* Following the causal graph in Figure 2 (c), we design a straightforward yet effective item predictor $\theta_{\text{item}}$ to estimate the effects of visibility bias by predicting interactions based on multimodal data of items only. Specifically, we first extract multimodal representations through latent encoders, and then concatenate them to feed into a predictor $\theta_{\text{item}}$. In this paper, we adopt a simple MLP with the LeakyReLU activation function [8] to predict $\hat{y}_i$. We also perform the BPR loss to supervise the training of the item predictor, defined as $\mathcal{L}_{\text{BPR}_i} = f_{\text{BPR}}(\hat{y}_{i_p}, \hat{y}_{i_n})$. During the training stage, we combine the BPR loss components from the multimodal recommender and item predictor together with a balance parameter $\alpha_2$ (the value is set as Table 4): $\mathcal{L}_{\text{BPR}} = \mathcal{L}_{\text{BPR}_{u,i}} + \alpha_2 \mathcal{L}_{\text{BPR}_i}$. During the inference stage, we follow the counterfactual inference

**Table 1: Statistics of multimodal datasets, with multimodal content Visual (V), Acoustic (A) and Textual (T).**

| Dataset | #Users | #Items | #Interactions | Modalities | Sparsity |
|---------|--------|--------|---------------|------------|----------|
| Baby | 19,445 | 7,050 | 139,110 | V, T | 99.899% |
| Tiktok | 9,319 | 6,710 | 59,541 | V, A, T | 99.904% |
| Allrecipes | 19,805 | 10,067 | 58,922 | V, T | 99.970% |

in Eq. 10 to adjust the predictions by eliminating NDE from TE:

$$y_{u,i} = \hat{y}_{u,i} * \text{sigm}(\hat{y}_i) - \gamma * \text{sigm}(\hat{y}_i), \tag{17}$$

where $\gamma$ represents the value of $\hat{y}_{u,i^*}$ in the counterfactual scenario, which is usually chosen empirically [45]. As a result, we can effectively alleviate the negative effects of visibility bias and generate accurate and fair recommendations.

## 4.3 Optimization

The overall objective of MoDiCF is to jointly optimize the MDDC module and the CFMR module. The overall objective is

$$\mathcal{L} = \mathcal{L}_{\text{diff}} + \mathcal{L}_{\text{BPR}} + \lambda_1 \mathcal{L}_{\text{CL}} + \lambda_2 \|\Theta\|_2^2, \tag{18}$$

where $\lambda_1$ and $\lambda_2$ are trade-off parameters, whose values are specified in Table 4. The last term is a regularization term to prevent overfitting with a small decay coefficient $\lambda_2 = 10^{-5}$. Considering the diffusion process usually requires more training iterations than recommender systems, we implement a two-stage training strategy to optimize the MoDiCF framework. In the first stage, we pre-train the MDDC module with the loss $\mathcal{L}_{\text{diff}}$. In the second stage, we jointly optimize the entire framework using $\mathcal{L}$. The algorithm of MoDiCF is summarized in Algorithm 1 in the appendix.

## 5 EXPERIMENTS

### 5.1 Data Preparation

We conduct extensive experiments on three publicly available multimodal recommendation datasets: Amazon Baby (Baby for short), Tiktok and Allrecipes[1]. The baby dataset contains user reviews and ratings on baby products, and each rating is considered a user-item interaction. It has visual and textual features with 4,096 and 1,024 dimensions. The Tiktok dataset contains visual, acoustic and textual features with dimensions of 128, 128 and 768, respectively. The Allrecipes dataset includes food recipes with visual and textual features with dimensions of 2,048 and 20, respectively. We follow the same partition settings in [46] for preparing the training, validation and test sets. A summary of dataset statistics is in Table 1.

Based on these, we construct incomplete multimodal datasets using the following strategy. First, we define a missing rate $MR = \frac{\sum_{m=1}^{M} |\mathcal{E}_{\text{ms}}^m|}{N_I \times M}$ to control the incompleteness level. To ensure that each item has at least one observed modality, $MR$ ranges from $(0, \frac{M-1}{M}]$. We set $MR = 0.4$ for all datasets unless otherwise specified. Modalities of each item are randomly dropped according to these rules. This incomplete multimodal content is used consistently across the training, validation and test sets to prevent information leakage.

### 5.2 Experimental Settings

*5.2.1 Baselines.* To evaluate the recommendation performance in incomplete multimodal scenarios, we adopt 13 representative

[1]https://github.com/HKUDS/MMSSL/tree/main

and/or state-of-the-art baseline methods from three classes. 1) Unimodal RS methods: **LightGCN** [12] and **AutoCF** [51]; 2) MM-Rec methods: **FREEDOM** [60], **MMSSL** [46], **BM3** [61], **MG** [57], **MCLN** [17], **MCDRec** [29], **DiffMM** [14] and **MDB** [35]; 3) Incomplete MMRec methods: **LRMM** [41], **CI²MG** [20] and **MILK** [1]. These methods are carefully selected to cover a wide range of aspects, including diffusion-based [14, 29], causal-based [17, 35], fairness-aware methods [35] and incomplete MMRec [1, 20, 41]. Note that some classic methods, e.g., VBPR [11] and MMGCN [48], are not included for comparison since the chosen methods such as MMSSL have been shown to outperform them in prior studies [14, 46]. For methods that cannot handle incomplete data, we follow [1] to use the mean strategy to impute missing modalities.

*5.2.2 Evaluation Metrics.* The goal of this work is to generate accurate and fair recommendations in incomplete multimodal scenarios. To this end, we evaluate all methods from the following aspects: 1) recommendation accuracy, 2) fairness, and 3) the combination of them. For recommendation accuracy, we adopt three widely-used metrics: Recall@$K$, Precision@$K$ and NDCG@$K$. To assess the visibility bias, inspired by the isolation index [43], we design a new metric, $F@K$. It measures the disparity between the proportion of incomplete items in the top-$K$ recommendations ($P_r@K$) and in the entire dataset ($P_d$), defined as

$$F@K = 1 - \left| \frac{P_r@K - P_d}{P_d} \right|, \quad P_r@K = \frac{\#\text{incomplete items}}{K}. \tag{19}$$

A higher $F@K$ indicates more fair exposure of incomplete items. Moreover, inspired by the harmonic mean in F1-score, we propose $F_{\text{fuse}}@K$ to appropriately combine accuracy and fairness:

$$F_{\text{fuse}}@K = \frac{2 \cdot F@K \cdot Precision@K}{F@K + Precision@K}. \tag{20}$$

All metrics are reported for $K = 10$ and $K = 20$. Following [42], we perform a paired t-test with $p < 0.05$ for significance test.

*5.2.3 Parameter Settings.* For a fair comparison, we first set the parameters of all baselines using the settings reported in their original papers and then fine-tune them for optimal performance on the validation set. The key hyperparameters of each baseline are outlined in Section C.1 in the appendix. Our method involves two training stages. In the pretraining stage, we start with a learning rate $lr$ of $10^{-4}$, using a scheduler to reduce it by a factor of 0.95 every 100 epochs for fast convergence. In the second stage, we use a constant $lr$ of $10^{-4}$ with a maximum epoch of 250. We empirically set the sampling step $T_s$ to 10. The trade-off paramter $\lambda_1$ is searched within $\{n \times 0.01\}_{n=1}^{20}$, while $\lambda_2$ is fixed at $10^{-5}$. The hyperparameters $\alpha_1, \alpha_2, \eta, \delta$ are tuned within $\mathcal{S} = \{n \times 0.1\}_{n=1}^{10}$, and the counterfactual coefficient $\gamma$ is searched within $\{10^{-n}\}_{n=0}^{3} \cup \{n \times 10\}_{n=1}^{5}$. In the multimodal recommender sub-module, we empirically set the number of layers $L = 2$ and tune the number of heads $H$ within $\{2^n\}_{n=0}^{4}$. To be fair, the embedding dimensions for all methods are set to 256, 256 and 128 for the Baby, Tiktok and Allrecipes datasets, respectively. Our MoDiCF framework is implemented in PyTorch and optimized using the Adam optimizer [15]. All experiments are conducted on a server with an AMD EPYC 9351P CPU, 2 NVIDIA L4 GPUs and 192GB RAM. To ensure reliable results, we repeat the experiments 10 times with different sampled incomplete datasets and model

Table 2: Comparison of accuracy and fairness performance (%) with baselines on the Baby, Tiktok and Allrecipes datasets. The best results are highlighted in bold, and the second best are underlined. $^*$ indicates the improvement is significant with $p < 0.05$. Note that, since unimodal RSs are not affected by visibility bias, they can be regarded as the ideal fairness reference.

| Dataset | Methods | Recall | | Precision | | NDCG | | $F$ | | $F_{\text{fuse}}$ | |
|---------|---------|--------|--------|-----------|--------|------|------|-----|-----|-------|-------|
| | | K=10 | K=20 | K=10 | K=20 | K=10 | K=20 | K=10 | K=20 | K=10 | K=20 |
| Baby | LightGCN | 4.24 ± 0.08 | 6.70 ± 0.22 | 0.45 ± 0.01 | 0.35 ± 0.01 | 2.27 ± 0.05 | 2.91 ± 0.09 | 87.44 ± 6.39 | 91.61 ± 3.16 | 0.80 ± 0.02 | 0.67 ± 0.02 |
| | AutoCF | 4.64 ± 0.10 | 6.87 ± 0.14 | 0.49 ± 0.01 | 0.37 ± 0.01 | 2.53 ± 0.06 | 3.10 ± 0.06 | 87.79 ± 0.54 | 91.15 ± 0.41 | 0.98 ± 0.02 | 0.73 ± 0.01 |
| | FREEDOM | 4.60 ± 0.62 | 7.51 ± 0.94 | 0.48 ± 0.06 | 0.40 ± 0.05 | 2.41 ± 0.32 | 3.15 ± 0.40 | 84.44 ± 1.36 | 88.65 ± 1.64 | 0.97 ± 0.13 | 0.79 ± 0.10 |
| | MMSSL | 5.11 ± 0.71 | 8.18 ± 1.10 | 0.54 ± 0.08 | 0.43 ± 0.06 | 2.76 ± 0.41 | 3.56 ± 0.50 | 85.12 ± 1.86 | 88.29 ± 1.66 | 0.97 ± 0.14 | 0.82 ± 0.11 |
| | BM3 | 4.99 ± 0.18 | 8.01 ± 0.32 | 0.52 ± 0.02 | 0.42 ± 0.02 | 2.62 ± 0.10 | 3.39 ± 0.14 | 84.50 ± 4.05 | 88.53 ± 1.94 | 1.04 ± 0.04 | 0.84 ± 0.03 |
| | MG | 5.10 ± 0.22 | 8.22 ± 0.28 | 0.54 ± 0.02 | 0.43 ± 0.02 | 2.69 ± 0.11 | 3.48 ± 0.13 | 84.38 ± 3.94 | 88.74 ± 1.81 | 1.07 ± 0.05 | 0.86 ± 0.03 |
| | MCLN | 4.79 ± 0.05 | 7.94 ± 0.14 | 0.50 ± 0.01 | 0.42 ± 0.01 | 2.42 ± 0.03 | 3.22 ± 0.04 | 84.22 ± 4.91 | 87.88 ± 3.53 | 1.00 ± 0.01 | 0.83 ± 0.01 |
| | MCDRec | 4.04 ± 0.14 | 6.67 ± 0.27 | 0.43 ± 0.01 | 0.35 ± 0.01 | 2.03 ± 0.08 | 2.70 ± 0.11 | 84.39 ± 4.23 | 88.46 ± 1.76 | 0.85 ± 0.03 | 0.70 ± 0.03 |
| | DiffMM | 5.11 ± 0.37 | 8.24 ± 0.52 | 0.54 ± 0.04 | 0.43 ± 0.03 | 2.67 ± 0.20 | 3.47 ± 0.23 | 85.53 ± 2.47 | 89.21 ± 0.52 | 1.07 ± 0.08 | 0.87 ± 0.05 |
| | MDB | 3.51 ± 0.12 | 5.81 ± 0.11 | 0.37 ± 0.01 | 0.31 ± 0.01 | 1.79 ± 0.06 | 2.38 ± 0.06 | 85.65 ± 1.92 | 89.01 ± 1.05 | 0.74 ± 0.03 | 0.62 ± 0.01 |
| | LRMM | 2.79 ± 0.11 | 4.34 ± 0.12 | 0.30 ± 0.01 | 0.24 ± 0.01 | 1.52 ± 0.07 | 1.92 ± 0.06 | 70.48 ± 1.88 | 72.96 ± 1.79 | 0.60 ± 0.02 | 0.47 ± 0.01 |
| | CI²MG | 5.13 ± 0.44 | 8.29 ± 0.64 | 0.54 ± 0.05 | 0.44 ± 0.03 | 2.76 ± 0.23 | 3.58 ± 0.28 | 85.68 ± 1.42 | 89.00 ± 0.99 | 1.08 ± 0.09 | 0.87 ± 0.07 |
| | MILK | 1.87 ± 0.04 | 3.38 ± 0.11 | 0.12 ± 0.01 | 0.10 ± 0.01 | 0.80 ± 0.03 | 1.10 ± 0.04 | 56.88 ± 6.05 | 68.53 ± 5.17 | 0.24 ± 0.01 | 0.20 ± 0.01 |
| | MoDiCF | 5.51 ± 0.17* | 8.76 ± 0.21* | 0.58 ± 0.02* | 0.46 ± 0.01* | 2.95 ± 0.10* | 3.78 ± 0.10* | 87.24 ± 1.09* | 90.12 ± 0.90* | 1.16 ± 0.03* | 0.92 ± 0.02* |
| Tiktok | LightGCN | 3.57 ± 1.39 | 5.35 ± 2.02 | 0.36 ± 0.14 | 0.27 ± 0.10 | 1.83 ± 0.72 | 2.28 ± 0.88 | 88.57 ± 0.78 | 92.23 ± 0.55 | 0.64 ± 0.25 | 0.51 ± 0.19 |
| | AutoCF | 3.55 ± 0.53 | 5.72 ± 0.68 | 0.36 ± 0.05 | 0.29 ± 0.03 | 1.79 ± 0.30 | 2.34 ± 0.34 | 88.55 ± 0.43 | 92.25 ± 0.34 | 0.71 ± 0.10 | 0.57 ± 0.07 |
| | FREEDOM | 5.07 ± 0.25 | 7.48 ± 0.37 | 0.51 ± 0.03 | 0.38 ± 0.02 | 2.53 ± 0.19 | 3.13 ± 0.22 | 85.64 ± 6.03 | 88.36 ± 3.62 | 1.01 ± 0.05 | 0.75 ± 0.04 |
| | MMSSL | 4.76 ± 0.23 | 7.38 ± 0.35 | 0.48 ± 0.02 | 0.37 ± 0.02 | 2.58 ± 0.16 | 3.24 ± 0.20 | 87.44 ± 0.97 | 90.05 ± 0.70 | 0.88 ± 0.07 | 0.71 ± 0.04 |
| | BM3 | 5.02 ± 0.33 | 7.59 ± 0.57 | 0.50 ± 0.03 | 0.38 ± 0.03 | 2.53 ± 0.22 | 3.18 ± 0.28 | 85.53 ± 4.23 | 88.46 ± 4.42 | 1.00 ± 0.07 | 0.76 ± 0.06 |
| | MG | 5.01 ± 0.29 | 7.71 ± 0.35 | 0.50 ± 0.03 | 0.39 ± 0.02 | 2.49 ± 0.08 | 3.16 ± 0.09 | 82.10 ± 6.55 | 88.09 ± 3.41 | 1.00 ± 0.06 | 0.77 ± 0.03 |
| | MCLN | 4.56 ± 0.46 | 7.25 ± 0.92 | 0.46 ± 0.05 | 0.36 ± 0.05 | 2.22 ± 0.34 | 2.89 ± 0.36 | 84.92 ± 8.50 | 88.18 ± 3.85 | 0.91 ± 0.09 | 0.72 ± 0.09 |
| | MCDRec | 4.39 ± 0.51 | 7.07 ± 0.90 | 0.44 ± 0.05 | 0.35 ± 0.05 | 2.28 ± 0.31 | 2.96 ± 0.36 | 84.72 ± 6.78 | 89.16 ± 3.19 | 0.85 ± 0.14 | 0.68 ± 0.13 |
| | DiffMM | 4.73 ± 0.66 | 7.60 ± 0.94 | 0.47 ± 0.07 | 0.38 ± 0.05 | 2.68 ± 0.31 | 3.40 ± 0.32 | 85.50 ± 4.49 | 89.68 ± 2.00 | 0.94 ± 0.13 | 0.76 ± 0.09 |
| | MDB | 4.27 ± 0.48 | 6.77 ± 0.46 | 0.43 ± 0.05 | 0.34 ± 0.02 | 2.16 ± 0.21 | 2.79 ± 0.15 | 84.56 ± 9.45 | 89.63 ± 4.91 | 0.85 ± 0.09 | 0.68 ± 0.04 |
| | LRMM | 3.27 ± 0.36 | 4.95 ± 0.56 | 0.33 ± 0.04 | 0.25 ± 0.03 | 1.70 ± 0.27 | 2.12 ± 0.28 | 80.27 ± 4.44 | 82.48 ± 3.20 | 0.65 ± 0.07 | 0.49 ± 0.06 |
| | CI²MG | 4.91 ± 0.39 | 7.74 ± 0.28 | 0.49 ± 0.04 | 0.39 ± 0.01 | 2.67 ± 0.21 | 3.38 ± 0.19 | 84.28 ± 1.08 | 91.32 ± 0.29 | 0.97 ± 0.09 | 0.76 ± 0.04 |
| | MILK | 1.70 ± 0.19 | 3.00 ± 0.33 | 0.10 ± 0.02 | 0.09 ± 0.01 | 0.68 ± 0.10 | 0.93 ± 0.11 | 70.49 ± 6.80 | 76.31 ± 5.84 | 0.20 ± 0.03 | 0.17 ± 0.02 |
| | MoDiCF | 5.94 ± 0.35* | 9.29 ± 0.54* | 0.59 ± 0.04* | 0.46 ± 0.03* | 3.15 ± 0.31* | 3.99 ± 0.30* | 88.15 ± 1.60* | 92.27 ± 1.32* | 1.18 ± 0.07* | 0.92 ± 0.05* |
| Allrecipes | LightGCN | 1.27 ± 0.10 | 2.12 ± 0.24 | 0.13 ± 0.01 | 0.11 ± 0.01 | 0.63 ± 0.06 | 0.84 ± 0.09 | 91.82 ± 3.70 | 92.65 ± 1.51 | 0.25 ± 0.02 | 0.21 ± 0.02 |
| | AutoCF | 1.17 ± 0.13 | 2.12 ± 0.14 | 0.12 ± 0.01 | 0.11 ± 0.01 | 0.57 ± 0.09 | 0.81 ± 0.07 | 92.38 ± 5.29 | 92.66 ± 2.61 | 0.23 ± 0.03 | 0.21 ± 0.01 |
| | FREEDOM | 1.07 ± 0.24 | 2.01 ± 0.34 | 0.11 ± 0.02 | 0.10 ± 0.02 | 0.54 ± 0.10 | 0.77 ± 0.12 | 85.05 ± 3.64 | 88.98 ± 2.89 | 0.21 ± 0.05 | 0.20 ± 0.03 |
| | MMSSL | 2.22 ± 0.35 | 3.18 ± 0.32 | 0.22 ± 0.04 | 0.16 ± 0.02 | 1.03 ± 0.16 | 1.27 ± 0.14 | 86.80 ± 7.23 | 87.26 ± 5.14 | 0.40 ± 0.07 | 0.30 ± 0.03 |
| | BM3 | 1.72 ± 0.43 | 2.80 ± 0.57 | 0.17 ± 0.04 | 0.14 ± 0.03 | 0.87 ± 0.25 | 1.15 ± 0.27 | 86.77 ± 7.26 | 89.21 ± 4.14 | 0.34 ± 0.09 | 0.28 ± 0.06 |
| | MG | 1.58 ± 0.39 | 2.64 ± 0.63 | 0.16 ± 0.04 | 0.13 ± 0.03 | 0.80 ± 0.19 | 1.07 ± 0.24 | 85.91 ± 6.93 | 89.36 ± 4.64 | 0.31 ± 0.08 | 0.26 ± 0.06 |
| | MCLN | 2.18 ± 0.38 | 3.08 ± 0.45 | 0.22 ± 0.04 | 0.15 ± 0.02 | 1.03 ± 0.19 | 1.25 ± 0.18 | 87.24 ± 9.45 | 86.37 ± 6.61 | 0.43 ± 0.08 | 0.31 ± 0.05 |
| | MCDRec | 1.96 ± 0.35 | 3.11 ± 0.50 | 0.20 ± 0.04 | 0.16 ± 0.03 | 1.00 ± 0.16 | 1.29 ± 0.19 | 85.54 ± 7.66 | 87.90 ± 5.19 | 0.39 ± 0.07 | 0.31 ± 0.05 |
| | DiffMM | 2.07 ± 0.28 | 3.05 ± 0.24 | 0.21 ± 0.03 | 0.15 ± 0.01 | 0.96 ± 0.07 | 1.21 ± 0.08 | 87.56 ± 7.20 | 88.76 ± 4.07 | 0.41 ± 0.06 | 0.30 ± 0.02 |
| | MDB | 1.91 ± 0.34 | 3.03 ± 0.19 | 0.19 ± 0.03 | 0.15 ± 0.01 | 0.90 ± 0.17 | 1.18 ± 0.13 | 88.71 ± 8.87 | 87.23 ± 4.28 | 0.38 ± 0.07 | 0.30 ± 0.02 |
| | LRMM | 1.03 ± 0.23 | 1.89 ± 0.29 | 0.10 ± 0.02 | 0.09 ± 0.01 | 0.50 ± 0.12 | 0.71 ± 0.12 | 57.57 ± 6.67 | 60.48 ± 6.38 | 0.21 ± 0.05 | 0.19 ± 0.03 |
| | CI²MG | 1.86 ± 0.34 | 3.23 ± 0.46 | 0.19 ± 0.03 | 0.16 ± 0.02 | 0.87 ± 0.15 | 1.21 ± 0.19 | 88.36 ± 3.00 | 89.42 ± 2.12 | 0.35 ± 0.07 | 0.32 ± 0.05 |
| | MILK | 0.69 ± 0.12 | 1.08 ± 0.14 | 0.05 ± 0.01 | 0.03 ± 0.01 | 0.35 ± 0.10 | 0.43 ± 0.10 | 54.16 ± 5.05 | 59.20 ± 4.27 | 0.09 ± 0.03 | 0.07 ± 0.01 |
| | MoDiCF | 2.56 ± 0.11* | 3.65 ± 0.28* | 0.26 ± 0.01* | 0.18 ± 0.01* | 1.23 ± 0.13* | 1.50 ± 0.18* | 94.12 ± 6.01* | 92.21 ± 5.45* | 0.51 ± 0.02* | 0.36 ± 0.03* |

initialization, and report average results and standard deviations. Other implementation details are provided in the appendix.

## 5.3 Performance Comparison

The average performance and standard deviations of each method on three datasets, in terms of both accuracy and fairness, are presented in Table 2. Since unimodal RSs, i.e., LightGCN and AutoCF, do not learn from multimodal data, they are naturally immune to visibility bias. Thus, we use their $F$ scores as a reference for ideal fairness. From these results, we observe the following: 1) The proposed MoDiCF method consistently outperforms all baselines in both accuracy and fairness on these datasets. For instance, on the Tiktok dataset, MoDiCF shows a 19.6% improvement in Recall@20. Meanwhile, its fairness score closely aligns with reference scores, indicating its effectiveness in addressing visibility bias. Note that mitigating visibility bias could help enhance accuracy; as more incomplete items are fairly treated, more accurate user-item preference learning can be achieved. 2) Although unimodal RSs are unaffected by visibility bias, they show limited accuracy as they lack the capability to exploit multimodal information. 3) MMRec methods

generally outperform unimodal RS. However, as they are not specifically designed for incomplete scenarios, they are still vulnerable to both data incompleteness and visibility bias. Among them, MCDRec and MDB are relatively more sensitive to these issues, thus showing limited performance. 4) Incomplete MMRec methods can partially handle data incompleteness but still exhibit several drawbacks. For example, LRMM shows limited performance as the simple autoencoders it uses are not sufficient to capture modality distributions for high-quality data completion. Meanwhile, as MILK is specially devised for new item recommendations and can only learn from complete items, it struggles in our setting where most training and test items are incomplete. CI²MG shows better accuracy but cannot properly handle visibility bias, leading to suboptimal fairness.

We further test the performance of several representative methods on three datasets across varying MRs with the metric $F_{\text{fuse}}$@20. As shown in Figure 4, MoDiCF consistently outperforms all baselines, indicating its effectiveness and robustness across various incomplete scenarios. CI²MG and DiffMM show competitive performance, while FREEDOM and MMSSL exhibit less stability, especially on the Tiktok dataset.

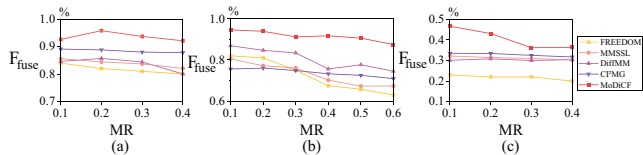

**Figure 4: Comparison with different MRs.**

**Table 3: Comparison of MoDiCF with its variants.**

| Variants | Baby | | | Tiktok | | |
|---|---|---|---|---|---|---|
| | Recall | $F$ | $F_{\text{fuse}}$ | Recall | $F$ | $F_{\text{fuse}}$ |
| MoDiCF-D+M | 8.39 | 89.69 | 0.89 | 7.70 | 91.54 | 0.77 |
| MoDiCF-D+Z | 8.32 | 86.99 | 0.88 | 7.18 | 90.36 | 0.72 |
| MoDiCF-D+R | 7.56 | 89.32 | 0.80 | 6.48 | 90.15 | 0.65 |
| MoDiCF-D+N | 8.38 | 89.80 | 0.88 | 7.23 | 91.13 | 0.72 |
| MoDiCF-con | 7.50 | 89.57 | 0.79 | 7.68 | 89.39 | 0.76 |
| MoDiCF-C | 8.42 | 87.58 | 0.89 | 8.66 | 89.71 | 0.86 |
| MoDiCF-D-C+M | 8.08 | 86.39 | 0.85 | 7.59 | 88.49 | 0.75 |
| MoDiCF | **8.76** | **90.12** | **0.92** | **9.29** | **92.27** | **0.92** |

## 5.4 Ablation Study

To validate the effectiveness of each component in MoDiCF, we conduct an ablation study by comparing MoDiCF with its variants, including 1) MoDiCF-D+{M, Z, R, N}: removes MDDC module and fills missing data with imputation strategies of means, zeros, random values, and nearest neighbors [25]; 2) MoDiCF-con: removes the modality-aware conditions from MDDC; 3) MoDiCF-C: removes counterfactual inference and only relies on the multimodal recommender for recommendations; 4) MoDiCF-D-C+M: replaces MDDC with a mean strategy and removes counterfactual inference. Due to space limitations, we only present results on Baby and Tiktok datasets with $K = 20$. Full results are available in the appendix.

As summarized in Table 3, the key findings are: 1) The MDDC module matters in addressing incompleteness, as its exclusion leads to a notable performance decline. Meanwhile, comparing the results of MoDiCF-C with existing incomplete MMRec methods further confirms the superiority of this module; 2) Counterfactual inference mechanism greatly contributes to the fairness of recommendations. Without it, there is a sharp drop in $F$ scores; 3) As our two modules are designed to complement each other, they work collaboratively to improve both accuracy and fairness. Removing either of them causes a clear performance drop in both aspects. 4) Overall, MoDiCF achieves the best results, indicating the effectiveness of its design.

## 5.5 Parameter Analysis

In this section, we analyze the impact of key hyperparameters on the performance of MoDiCF, including the trade-off parameters $\lambda_1$ and $\lambda_2$, the diffusion sampling step $T_s$ and the counterfactual coefficient $\gamma$. We conduct experiments on three datasets and report the results of an accuracy-fairness combined metric $F_{\text{fuse}}@20$ in Figure 5. Additional parameter analysis is provided in the appendix.

**Impact of trade-off parameters.** We respectively vary the values of $\lambda_1$ and $\lambda_2$ within the ranges of $\{n \times 0.01\}_{n=1}^{20}$ and $\{10^{-n}\}_{n=0}^{6}$. As shown in Figures 5 (a) and (b), the optimal values of $\lambda_1$ and $\lambda_2$ are around 0.7 and $10^{-5}$, respectively, while larger values of $\lambda_1$ and $\lambda_2$ usually lead to a decreased and unstable performance.

**Impact of sampling steps.** The sampling step $T_s$ is a key hyperparameter as it directly affects the quality of data generation. Following the strategy in [37], using a smaller $T_s$ becomes possible

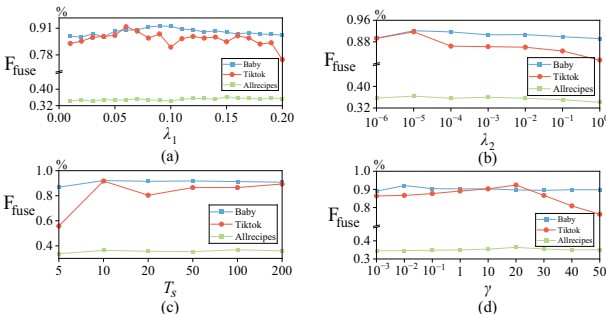

**Figure 5: Impacts of the trade-off parameters (a) $\lambda_1$, (b) $\lambda_2$, (c) sampling step $T_s$ and (d) $\gamma$ on the performance of MoDiCF.**

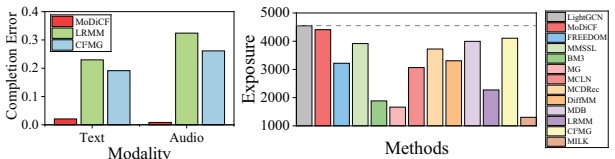

**Figure 6: A case study of an item in the Tiktok dataset.**

to reduce computational complexity while maintaining high generation quality. Here, we vary $T_s$ within the set $\{5, 10, 20, 50, 100, 200\}$. As shown in Figure 5 (c), the performance of MoDiCF becomes relatively stable when $T_s \geq 10$, while an extremely small $T_s$ may lead to inaccurate data generation and clear performance decline. Hence, we set $T_s = 10$ in our experiments.

**Impact of the counterfactual coefficient.** The counterfactual coefficient $\gamma$ controls the strength of counterfactual adjustments during recommendations. We vary $\gamma$ within $\{10^{-n}\}_{n=0}^{3} \cup \{n \times 10\}_{n=1}^{5}$ and observe that the optimal value is around $10^{-2}$, 20 and 20 for the Baby, Tiktok and Allrecipes datasets, as shown in Figure 5 (d). A larger $\gamma$ may result in over-intervention while a smaller $\gamma$ may fail to alleviate visibility bias, thus leading to a performance drop.

## 6 CASE STUDY

To illustrate how MoDiCF enhances data completion and mitigate visibility bias, we present a case study (Figure 6) using a randomly selected item with missing text and audio from Tiktok dataset. MoDiCF not only achieves the best data completion quality but also delivers fair exposure of the item, closely aligning with the unimodal method LightGCN [12]. See Appendix for more details.

## 7 CONCLUSION

In this paper, we focus on a prevalent yet challenging scenario in multimodal recommendations, where modality data is incomplete and the visibility bias is severe. To address these issues, we propose a novel Modality-Diffused Counterfactual (MoDiCF) framework for incomplete MMRec. With two specially devised modules, i.e., the modality-aware diffusion module and the counterfactual inference module, MoDiCF can effectively generate missing modalities from captured modality-specific distributions and alleviate visibility bias by counterfactual inference. Extensive experiments on three real-world datasets demonstrate the superiority of MoDiCF in terms of both recommendation accuracy and fairness and validate the specific design of MoDiCF. In the future, we plan to explore finer-grained fairness in MMRec.

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

## A  ALGORITHM

In this section, we summarize the proposed MoDiCF framework in Algorithm 1. It consists of two main stages during training: 1) the pretraining stage uses the loss function in Eq. 5 to pretrain the MDDC module, while 2) the second stage trains the entire MoDiCF framework using the whole loss function in Eq. 18. For a well-trained MoDiCF, we can effectively obtain complete multimodal data from the MDDC module and generate accurate and fair recommendations from the CFMR module.

## B  COMPUTATIONAL COMPLEXITY ANALYSIS

To evaluate the computational efficiency of MoDiCF, we compare the average running time of each training epoch with other representative methods on three datasets. As shown in Figure 7, MoDiCF

---

**Algorithm 1** MoDiCF: Modality-Diffused CounterFactual Framework

**Input:** User set $\mathcal{U}$, item set $\mathcal{I}$, user-item interaction matrix $\mathbf{Y}$, incomplete multimodal data $\{\mathcal{X}^m\}_{m=1}^M$, indicator matrix $\mathbf{E}$, parameters $d_m$, $lr$, $\lambda_1$, $\lambda_2$, $T_s$, $\gamma$, $\alpha_1$, $\alpha_2$, $\eta$, $\delta$, $L$ and $H$.
**Output:** Completed multimodal data $\{\hat{\mathcal{X}}^m\}_{m=1}^M$ and recommendation results.

  **Pretraining Stage:**
1: Initialize missing modalities $\mathcal{X}_{\mathrm{ms}}^m$ with zero vectors $\mathbf{0}$;
2: Parameter initialization for the MDDC module;
3: **while** not converged **do**
4:   Encode $\mathcal{X}_{\mathrm{ob}}^m$ into latent spaces as $\mathcal{V}_{\mathrm{ob}}^m$;
5:   Perform the forward process $q(\mathbf{v}_{\mathrm{ob},1:T}^m|\mathbf{v}_{\mathrm{ob},0}^m)$ via Eq. 1;
6:   Extract modality-aware conditions $\mathbf{v}_{\mathrm{cn},0}^m$;
7:   Perform the reverse process $p_{\theta_{\mathrm{cn}}}(\mathbf{v}_{\mathrm{ob},t-1}^m|\mathbf{v}_{\mathrm{ob},t}^m, \mathbf{v}_{\mathrm{cn},t}^m)$ with $\theta_{\mathrm{cn}}$ via Eq. 2;
8:   Perform the generation process for $\mathbf{v}_{\mathrm{ms}}^m$ via Eq. 8 and obtain completed data $\hat{\mathcal{X}}_{\mathrm{ms}}^m$ via latent decoder $D_{\mathrm{diff}}^m$;
9:   Update multimodal data by replacing $\mathcal{X}_{\mathrm{ms}}^m$ with $\hat{\mathcal{X}}_{\mathrm{ms}}^m$;
10:   Optimize the MDDC module with $\mathcal{L}_{\mathrm{diff}}$ in Eq. 5;
11: **end while**
  **Joint Training Stage:**
12: Parameter initialization for the CFMR module;
13: Construct the user-item graph $\mathcal{G}$ based on $\mathbf{Y}$;
14: **while** not converged **do**
15:   Perform steps 4-9 to train the MDDC module and generate completed multimodal data $\hat{\mathcal{X}}^m$;
16:   Extract latent representation $\mathcal{V}^m$ based on $\hat{\mathcal{X}}^m$;
17:   Extract modality-aware features $\tilde{\mathbf{v}}_u^m$ and $\tilde{\mathbf{v}}_i^m$ via Eq. 11;
18:   Build modality-aware interaction matrix $\mathbf{Y}^m$ based on $\mathbf{Y}_{u,i}^m = \mathrm{sim}(\tilde{\mathbf{v}}_u^m, \tilde{\mathbf{v}}_i^m)$;
19:   Learn modality-aware embeddings via Eqs. 12, 13 and 14;
20:   Fuse modality-aware ID embeddings and user, item features to obtain $\mathbf{f}_u$ and $\mathbf{f}_i$;
21:   Compute $\mathcal{L}_{\mathrm{CL}}$ and $\mathcal{L}_{\mathrm{BPR}_{u,i}}$ from the multimodal recommender via Eqs. 15 and 16;
22:   Compute $\mathcal{L}_{\mathrm{BPR}_i}$ from the item predictor;
23:   Optimize the entire framework with $\mathcal{L}$ in Eq. 18;
24: **end while**
  **Inference Stage:**
25: Generate predictions $\hat{y}_{u,i}$ from multimodal recommender;
26: Generate predictions $\hat{y}_i$ from item predictor;
27: Perform counterfactual inference to obtain adjusted ranking scores $y_{u,i}$ via Eq. 17;
28: **return** Recommendation results based on $y_{u,i}$.

---

achieves a competitive running time compared to other methods. It is worth noting that MoDiCF involves a generation process during each training epoch for data completion. Therefore, it has relatively higher time costs than methods that do not perform data completion. However, MoDiCF still maintains reasonable efficiency. Notably, the running time of MoDiCF is close to and usually lower than $\mathrm{CI}^2\mathrm{MG}$. These results demonstrate the efficiency of MoDiCF in handling incomplete multimodal recommendation tasks.

**Figure 7: Comparison of running time of different methods on three datasets.**

## C ADDITIONAL EXPERIMENTAL DETAILS

### C.1 Baselines

We provide a detailed list of the baselines used in our experiments, including unimodal RS methods, MMRec methods, and incomplete MMRec methods. For these methods, we use a learning rate of 0.001 for training unless otherwise specified.

**1) Unimodal RS methods**

- **LightGCN** [12]: a classic unimodal RS method with simplified and efficient graph convolutional networks. We set the number of layers to 3 and the weight decay to $10^{-4}$.
- **AutoCF** [51]: a self-supervised learning method incorporating adaptive data augmentation for robust recommendations. We set the number of attention heads to 4 and the weight decay to $10^{-7}$.

**2) MMRec methods:**

- **FREEDOM** [60]: an efficient design for MMRec by reducing the complexity of graph structure learning. We set the numbers of GCN layers for the user-item graph and the item-item graph, respectively, to 2 and 1.
- **MMSSL** [46]: an advanced MMRec using adversarial augmentation and contrastive self-supervised learning. We use a learning rate of $5.5 \times 10^{-4}$, a weight decay regularizer of $10^{-5}$ and 2 layers of graph propagation.
- **BM3** [61]: a high-efficient method that reduces the need for auxiliary graph construction and negative sampling. In this method, the dropout rate for embedding perturbation is set to 0.3 and the regularization coefficient is set to 0.1.
- **MG** [57]: a robust method with a novel optimization process to reduce information adjustment risks and noise risks. We use BM3 as its backbone. The scaling coefficients $\alpha_1$ and $\alpha_2$ are set to 1 and 0.1, and the interval parameter $\beta$ is set to 3.
- **MCLN** [17]: a causal-based method to alleviate spurious correlations caused by irrelevant multimodal features. We use a decay coefficient of $10^{-3}$ for regularization and 5 layers for counterfactual learning.
- **MCDRec** [29]: a diffusion-based model to learn high-order multimodal knowledge in item embeddings. The number of graph convolutional layers is set to 2 and the loss coefficient $\lambda$ is $10^{-3}$.
- **DiffMM** [14]: a robust method with a graph diffusion model to reduce irrelevant noise caused by random augmentation. We set the number of GNN layers to 1 and the decay coefficient to $10^{-5}$. The loss weights for the diffusion module and the contrastive learning are respectively set to $10^{-1}$ and $10^{-2}$. The temperature coefficient for contrastive learning is set to 0.5.
- **MDB** [35]: a causal-based method to reduce the bias caused by dominently prevailing modalities. We follow [35] to use MMGCN

**Table 4: Hyperparameters of MoDiCF on three datasets.**

| Datasets | $d_m$ | $lr$ | $\lambda_1$ | $\lambda_2$ | $T_s$ | $\gamma$ | $\alpha_1$ | $\alpha_2$ | $\eta$ | $\delta$ | $L$ | $H$ |
|---|---|---|---|---|---|---|---|---|---|---|---|---|
| Baby | 128 | $10^{-4}$ | 0.09 | $10^{-5}$ | 10 | 0.01 | 1 | 0.7 | 0.7 | 0.4 | 2 | 8 |
| Tiktok | 128 | $10^{-4}$ | 0.06 | $10^{-5}$ | 10 | 20 | 0.7 | 0.3 | 0.6 | 0.3 | 2 | 4 |
| Allrecipes | 128 | $10^{-4}$ | 0.15 | $10^{-5}$ | 10 | 20 | 0.6 | 0.5 | 0.3 | 0.4 | 2 | 8 |

[48] as its backbone. The learning rate is set to 0.0005 and the debiasing strength is set to 0.5.

**3) Incomplete MMRec methods:**

- **LRMM** [41]: a classic method that uses autoencoders to recover incomplete multimodal data. The learning rate is set to $10^{-4}$.
- **CI²MG** [2] [20]: a method that uses clustering-based imputation to recover missing features and performs cross-modal transport to refine representations. We set the learning rate to $10^{-4}$. The loss weights $\lambda_1, \lambda_2, \lambda_3$ are respectively set to 1, 1 and $10^{-5}$.
- **MILK** [1]: a method based on invariant learning to learn features that remain invariant across different incomplete scenarios. We follow [1] to set the loss weights $\beta = 1000$ and $\lambda = 0.05$.

### C.2 Implementation Details

In this section, we specify the values of all the hyperparameters involved in MoDiCF in Table 4 on three datasets for higher reproducibility. These parameters are empirically chosen for optimal performance. Please refer to Sections 5.5 and D.2 for a detailed analysis of these hyperparameters.

## D ADDITIONAL RESULTS

### D.1 Ablation Study

Here, we provide the full results of the ablation study on three datasets, including Baby, Tiktok and Allrecipes. As shown in Table 5, the key findings are overall consistent with the summarized results in the main paper. Both the MDDC and CFMR modules are essential for accurate and fair recommendations, as the exclusion of either of them leads to a notable performance drop. MoDiCF achieves the best results, validating the effectiveness of its design.

### D.2 Parameter Analysis

Apart from the four key hyperparameters analyzed in Section 5.2.3, several other parameters exist in the proposed MoDiCF, including the balance parameters $\alpha_1$ and $\alpha_2$, fusion weights $\eta$ and $\delta$, the number of layers $L$, the number of the attention heads $H$ and the dimensionality $d_m$. Although they are not covered in the main paper due to space limitations, this section presents a series of experiments to offer a comprehensive analysis of them.

**Impact of balance parameters.** We vary the values of $\alpha_1$ and $\alpha_2$ within $\{n \times 0.1\}$ and have the performance recorded in Figures 8 (a) and (b). We can find that $\alpha_1$ and $\alpha_2$ are relatively more stable on the Baby and Allrecipes datasets, while they lead to a clear performance drop when $\alpha_1 > 0.7$ and $\alpha_2 > 0.4$ on Tiktok dataset.

**Impact of fusion weights.** For the fusion weights $\eta$ and $\delta$, we vary their values within a range of $\{n \times 0.1\}$. From Figures 8 (c) and (d), we can observe that a larger value of $\eta$ could lead to a performance decline, particularly on the Tiktok dataset. On the

---

[2] We thank the authors of [20] for providing us with the source code.

**Table 5: The full results of the ablation study.**

| Variants | Baby | | | | | | Tiktok | | | | | | Allrecipes | | | | | |
|---|---|---|---|---|---|---|---|---|---|---|---|---|---|---|---|---|---|---|
| | Recall | | F | | $F_{fuse}$ | | Recall | | F | | $F_{fuse}$ | | Recall | | F | | $F_{fuse}$ | |
| | K=10 | K=20 | K=10 | K=20 | K=10 | K=20 | K=10 | K=20 | K=10 | K=20 | K=10 | K=20 | K=10 | K=20 | K=10 | K=20 | K=10 | K=20 |
| MMoDiCF-D+M | 5.12 | 8.39 | 87.07 | 89.69 | 1.08 | 0.89 | 4.73 | 7.70 | 87.07 | 91.54 | 0.94 | 0.77 | 1.68 | 3.26 | 86.02 | 92.05 | 0.33 | 0.33 |
| MoDiCF-D+Z | 5.14 | 8.32 | 86.91 | 86.99 | 1.09 | 0.88 | 4.62 | 7.18 | 87.24 | 90.36 | 0.92 | 0.72 | 1.19 | 2.47 | 87.53 | 88.69 | 0.24 | 0.25 |
| MoDiCF-D+R | 4.75 | 7.56 | 86.17 | 89.32 | 1.01 | 0.80 | 3.62 | 6.48 | 85.28 | 90.15 | 0.72 | 0.65 | 1.92 | 2.86 | 88.09 | 91.91 | 0.38 | 0.29 |
| MoDiCF-D+N | 5.19 | 8.38 | 86.63 | 89.80 | 1.10 | 0.88 | 4.13 | 7.23 | 86.56 | 91.13 | 0.82 | 0.72 | 1.56 | 2.72 | 88.60 | 92.01 | 0.31 | 0.27 |
| MoDiCF-con | 4.67 | 7.50 | 81.91 | 89.57 | 0.98 | 0.79 | 3.82 | 7.68 | 85.32 | 89.39 | 0.76 | 0.76 | 2.33 | 3.22 | 89.39 | 91.79 | 0.46 | 0.32 |
| MoDiCF-C | 5.19 | 8.42 | 86.18 | 87.58 | 1.09 | 0.89 | 5.61 | 8.66 | 86.28 | 89.71 | 1.11 | 0.86 | 2.21 | 3.59 | 87.71 | 90.52 | 0.44 | 0.35 |
| MoDiCF-D-C+M | 4.95 | 8.08 | 85.97 | 86.39 | 1.04 | 0.85 | 4.70 | 7.59 | 85.40 | 88.49 | 0.93 | 0.75 | 2.15 | 3.25 | 84.05 | 89.04 | 0.43 | 0.32 |
| MoDiCF | **5.51** | **8.76** | **87.24** | **90.12** | **1.16** | **0.92** | **5.94** | **9.29** | **88.15** | **92.27** | **1.18** | **0.92** | **2.56** | **3.65** | **94.12** | **92.21** | **0.51** | **0.36** |

**Table 6: Comparison of various extensions of MoDiCF based on representative MMRec models.**

| Variants | Baby | | | | | | Tiktok | | | | | | Allrecipes | | | | | |
|---|---|---|---|---|---|---|---|---|---|---|---|---|---|---|---|---|---|---|
| | Recall | | F | | $F_{fuse}$ | | Recall | | F | | $F_{fuse}$ | | Recall | | F | | $F_{fuse}$ | |
| | K=10 | K=20 | K=10 | K=20 | K=10 | K=20 | K=10 | K=20 | K=10 | K=20 | K=10 | K=20 | K=10 | K=20 | K=10 | K=20 | K=10 | K=20 |
| MoDiCF | **5.51** | **8.76** | **87.24** | **90.12** | **1.16** | **0.92** | **5.94** | **9.29** | **88.15** | **92.27** | **1.18** | **0.92** | **2.56** | **3.65** | **94.12** | **92.21** | **0.51** | **0.36** |
| MG | 5.10 | 8.22 | 84.38 | 88.74 | 1.07 | 0.86 | 5.01 | 7.71 | 82.10 | 88.09 | 1.00 | 0.77 | 1.58 | 2.64 | 85.91 | 89.36 | 0.31 | 0.26 |
| MoDiCF w/ MG | 5.21 | 8.41 | 86.89 | 89.93 | 1.09 | 0.88 | 5.45 | 8.42 | 87.25 | 90.58 | 1.08 | 0.84 | 2.42 | 3.04 | 93.24 | 91.80 | 0.48 | 0.30 |
| DiffMM | 5.11 | 8.24 | 85.53 | 89.21 | 1.07 | 0.87 | 4.73 | 7.60 | 85.50 | 89.68 | 0.94 | 0.76 | 2.07 | 3.05 | 87.56 | 88.76 | 0.41 | 0.30 |
| MoDiCF w/ DiffMM | 5.40 | 8.52 | 86.30 | 89.90 | 1.13 | 0.89 | 5.79 | 9.09 | 86.80 | 89.63 | 1.15 | 0.90 | 2.47 | 3.21 | 93.53 | 89.78 | 0.49 | 0.32 |

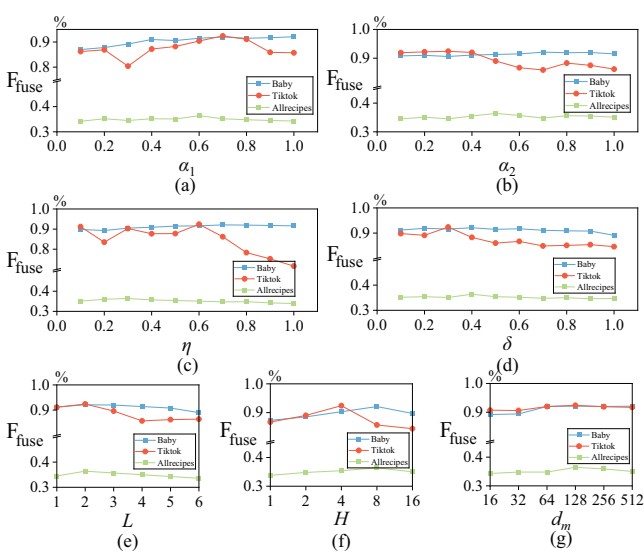

**Figure 8: Impacts of balance parameters (a) $\alpha_1$, (b) $\alpha_2$, fusion weights (c) $\eta$, (d) $\delta$, (e) the number of layers $L$, (f) the number of attention heads $H$ and (g) the dimensionality $d_m$.**

other hand, $\delta$ has a less significant impact on performance, with the optimal results occurring when $\delta$ is around 0.3.

**Impact of the number of layers.** We test the value of $L$ from 1 to 6 in increments of 1. As shown in Figure 8 (e), the optimal performance is achieved when $L = 2$. Beyond this point, the performance gradually declines.

**Impact of the number of attention heads.** We explore the number of attention heads, $H$, with values from the set $\{2^n\}_{n=0}^4$. The results in Figure 8 (f) show that the optimal performance occurs with $H$ set to 8, 4 and 8, respectively, for the Baby, Tiktok and Allrecipes datasets.

**Impact of the dimensionality.** We vary the dimensionality $d_m$ from $\{2^n\}_{n=4}^9$. As shown in Figure 8 (g), the performance becomes relatively stable when $d \geq 128$ on all three datasets. Therefore, we set $d_m = 128$ in our experiments.

## E  CASE STUDY

We randomly select an item from the Tiktok dataset with two missing modalities: text and audio modalities. We first evaluate the modality completion quality by comparing it with two incomplete MMRec methods, measuring the mean square error between the ground truth and the completed multimodal data. Note that, MILK is not comparable here, as it does not explicitly learn representations from incomplete data. It is evident from Figure 6 that, benefiting from the well-designed MDDC module, MoDiCF achieves the best data completion quality. Additionally, we compare the recommendation exposure of this item across different methods. Since the unimodal RS method LightGCN [12] is not affected by visibility bias, we use its exposure as the ideal fairness reference. As we can observe, MoDiCF shows the highest exposure for this incomplete item, closely aligning with the reference method LightGCN. These results indicate the effectiveness of our method in enhancing data completion and addressing visibility bias.

## F  EXTENSIBILITY OF MODICF

The proposed MoDiCF framework possesses strong extensibility, and it can be easily instantiated into various specific models by taking different MMRec models as the multimodal recommender. In this section, we present the comparison results of two MoDiCF variants based on two representative MMRec models: MG [57] and DiffMM [14]. As shown in Table 6, both variants demonstrate significant improvements in recommendation accuracy and fairness over their respective original models, validating the effectiveness of the MoDiCF framework design for handling incomplete MMRec.

