# OpenReview forum: "Generating with Fairness: A Modality-Diffused Counterfactual Framework for Incomplete Multimodal Recommendations"
_ACM.org/TheWebConf/2025/Conference — WWW 2025 Poster_

### Official Review · Reviewer_YRPU · 2024-11-25

**Novelty:** 5
**Technical Quality:** 4

**Review:**

The paper proposes a novel framework named MoDiCF, which addresses the challenges of data incompleteness and visibility bias in multimodal recommendation systems by designing a modality-diffused data completion module and a counterfactual multimodal recommendation module. Furthermore, the authors introduce fairness evaluation metrics and incorporate both accuracy and fairness into the experimental evaluations. The overall structure of the paper is clear, the experiments are relatively comprehensive, and the baseline comparisons are well-analyzed, demonstrating a good overall quality. This approach exhibits relatively strong originality in studying data incompleteness and fairness in multimodal recommendation systems.

However, the conclusion regarding "the prioritization of items’ multimodal data over user preference alignment" lacks detailed exploration and in-depth discussion.

Some technical elements, such as the mathematical notations in Section 4.2, are overly complex. Simplifying these notations or providing additional illustrative examples would enhance readability. Additionally, in line 548, the meaning of \hat{\bf e}_{u} is not defined or explained earlier in the text.

The rationale behind the design of the fairness evaluation metrics is inadequately explained.

In the experiments concerning the counterfactual coefficient, the observed large differences in the coefficient values between the Baby dataset and the other two datasets are not sufficiently analyzed or discussed.

In summary, the paper demonstrates good novelty, but there are issues with clarity in some sections and insufficient elaboration on a few critical points.

**Questions:**

The conclusion regarding "the prioritization of items' multimodal data over user preference alignment" lacks detailed exploration and in-depth discussion.

In the experiments concerning the counterfactual coefficient, the observed large differences in the coefficient values between the Baby dataset and the other two datasets are not sufficiently analyzed or discussed.

**Reviewer Confidence:**

3: The reviewer is confident but not certain that the evaluation is correct

**Scope:**

3: The work is somewhat relevant to the Web and to the track, and is of narrow interest to a sub-community

---

### Official Review · Reviewer_w9qJ · 2024-11-29

**Novelty:** 5
**Technical Quality:** 5

**Review:**

This paper tries to address the missing modality problem in multi-modality recommendation (MMRec) and the unfairness for the items with incomplete modalities caused by model training. The proposed MoDiCF framework generates missing modality data for incomplete items using the diffusion model and mitigates the negative causal effects of visibility bias using a counterfactual multimodal recommendation module.

## Strong points
1. Clear writing and enough related work.
2. The proposed model correlates with the proposed remaining challenge and motivations.
3. Comprehensive baselines and experiments. Nice looking charts and figures.
## Weakness
1. See question 1, more explanation is needed on the cause of the suppression of the items with incomplete modalities.
2. See question 2, further elaboration is needed in Section 4.2.2 to link with Equations 9 and 10.
3. See question 3, the new fairness metric proposed in Section 5.2.2 seems inappropriate.
4. Too much technical details in the title and introduction part. It seems like mixing five papers together, and it makes readers hard to imagine what this paper is about. Actually, the most important challenge is the modality missing problem, and most part of the model is focused on this problem.

**Questions:**

1. In the illustration for Figure 1, the decreasing item exposure of the items with missing modalities is attributed to the unfair treatment of these items. Could you please elaborate on why this is not caused by the intrinsic item quality difference between items with more or less modality data?
2. In Figure 3, it seems that the item predictor in the Counterfactual Multimodal Recommendation Module has the input of multi-modal contents with complete modalities, including the original existing modalities and the generated ones from the diffusion model. If all input items have complete modalities, how does the item predictor learn the Total Indirect Effect (TIE) caused by the missing modalities?
3. As for the evaluation metrics for fairness in Section 5.2.2, why define a new metric that only emphasizes the predicted score of incomplete items? Most fairness metrics focus on the discrepancies in the accuracy of the two groups, not the absolute predicted score. The latter method does not reflect whether the model treats different groups equally, instead, it only favors the model that gives certain groups higher scores.
4. It seems like the negative sampling strategy for evaluation is full sampling by the value of Precision and Recall? This choice is not mentioned in the experiment section.

**Reviewer Confidence:**

3: The reviewer is confident but not certain that the evaluation is correct

**Scope:**

4: The work is relevant to the Web and to the track, and is of broad interest to the community

---

### Official Review · Reviewer_7P8P · 2024-12-02

**Novelty:** 4
**Technical Quality:** 5

**Review:**

This paper proposes a modality-diffused counterfactual framework to address the issue of incomplete data in multimodal recommendations. The authors argue that: (1) Existing methods fail to effectively capture modality distributions to impute missing values. (2) Items with missing modalities tend to receive less exposure during recommendations, introducing bias. To tackle these issues, the authors incorporate conditional diffusion models to impute missing values, albeit with an additional computational overhead. Furthermore, they analyze visibility bias from a causal perspective and introduce metrics such as Total Effect (TE), Natural Direct Effect (NDE), and Total Indirect Effect (TIE) to mitigate it.

Strengths: (1) The paper is well-written and well-organized, making it easy to follow. The authors' effort in clear and concise writing is commendable. (2) Comprehensive experiments are conducted to verify the effectiveness of the proposed methods. (3) The paper provides detailed explanations of the algorithms and supplementary experiments beyond page limitations, aiding reviewers in understanding the contributions thoroughly.
Weaknesses:

Weaknesses: (1) Baseline Performance: A key concern is the lower baseline results compared to those reported in the original papers. Upon reviewing the original baseline results, the superiority of the proposed method seems no longer valid, but I require the authors' clarification to confirm this. (2) Evaluation Metrics: The inclusion of F_{fuse}@K in ablation studies and hyper-parameter analysis appears questionable. More details are elaborated below. (3) Method Adaptability: The proposed workflow is somewhat complex, requiring extensive hyper-parameter tuning for new datasets. This complexity may limit its practical adoption and impact on the recommendation systems community.

**Questions:**

(1) Baseline Performance in Table 2: I carefully reviewed the original papers for BM3, FREEDOM, MMSSL, DiffMM, and MILK. Most of the baseline results reported in your work are noticeably lower than those in the original papers. Notably, MILK performs significantly worse, even underperforming LightGCN and AutoCF, which seems unusual. Moreover, MoDiCF does not consistently outperform the baseline results as reported in the original papers. Considering that Baby, TikTok, and AllRecipes are widely used and standard datasets in multimodal recommendation research, are there any specific data processing steps or modifications to model architectures that might explain this discrepancy? Clear clarification on this point is crucial.

(2) Please specify which dataset is being used in Figure 4.

(3) It would improve clarity if you explicitly state that K=20 in Table 3.

(4) It is a bit complex to adapt this method to other tasks/datasets. It requires to train the diffusion model first before training the recommender. In addition, this method contains 12 hyperparameters to tune for a new task. I am a bit worried about the adaptation of this approach.

(5) Since this paper aims to address incomplete multimodal data in recommendation, I am curious about the missing rates. Could you include the missing rate for each modality and the overall missing rate for all items in Table 1?

(6) The abstract emphasizes that the primary goal of the proposed method is to enhance recommendation accuracy by addressing incomplete data, which negatively affects performance due to inaccurate imputation and visibility bias. However, the metric F_{fuse}@K appears to overemphasize fairness. For instance, if Method A achieves better accuracy than Method B but performs worse on the fairness-related metric like F value, Method B might still achieve a higher F_{fuse}@K. In this scenario, which method should be preferred? From my perspective, accuracy is the ultimate goal, and I would choose Method A. Thus, is it reasonable to report F_{fuse}@K in ablation studies and hyper-parameter analysis? Your case differs from some fairness-focused research, where fairness and accuracy are treated as two orthogonal objectives, and improving one often comes at the expense of the other.

**Reviewer Confidence:**

4: The reviewer is certain that the evaluation is correct and very familiar with the relevant literature

**Scope:**

4: The work is relevant to the Web and to the track, and is of broad interest to the community